# Comparison of Soft Indicator and Poisson Kriging for the Noise-Filtering and Downscaling of Areal Data: Application to Daily COVID-19 Incidence Rates

**Pierre Goovaerts** [1,*], **Thomas Hermans** [2], **Peter F. Goossens** [3] **and Ellen Van De Vijver** [3,4]

1   BioMedware, Inc. 167 Little lake dr., Ann Arbor, MI 48103, USA
2   Department of Geology, Ghent University, Campus Sterre, Krijgslaan 281, 9000 Ghent, Belgium;
    thomas.hermans@ugent.be
3   Department of Environment, Ghent University, Campus Coupure, Coupure Links 653, 9000 Ghent, Belgium;
    pfrgooss.goossens@ugent.be (P.F.G.); ervdevij.vandevijver@ugent.be (E.V.D.V.)
4   Department of Environment and Spatial Development, Government of Flanders, Koning Albert II-laan 20
    bus 8, 1000 Brussels, Belgium
*   Correspondence: goovaerts@biomedware.com

**Abstract:** This paper addresses two common challenges in analyzing spatial epidemiological data, specifically disease incidence rates recorded over small areas: filtering noise caused by small local population sizes and deriving estimates at different spatial scales. Geostatistical techniques, including Poisson kriging (PK), have been used to address these issues by accounting for spatial correlation patterns and neighboring observations in smoothing and changing spatial support. However, PK has a limitation in that it can generate unrealistic rates that are either negative or greater than 100%. To overcome this limitation, an alternative method that relies on soft indicator kriging (IK) is presented. The performance of this method is compared to PK using daily COVID-19 incidence rates recorded in 2020–2021 for each of the 581 municipalities in Belgium. Both approaches are used to derive noise-filtered incidence rates for four different dates of the pandemic at the municipality level and at the nodes of a 1 km spacing grid covering the country. The IK approach has several attractive features: (1) the lack of negative kriging estimates, (2) the smaller smoothing effect, and (3) the better agreement with observed municipality-level rates after aggregation, in particular when the original rate was zero.

**Keywords:** COVID-19; Belgium; Poisson kriging; indicator kriging; disaggregation; semivariogram

## 1. Introduction

Geostatistics has emerged as a powerful tool for the analysis of spatially correlated data in medical geography [1]; in particular, disease incidence and mortality rates. These rates are typically recorded at the level of administrative units, such as census tracts, counties or municipalities which can be sparsely populated and of different sizes and shapes. These areal data tend to be noisy when recorded for small population sizes, while the aggregation process may distort the true underlying spatial patterns of the disease and the subsequent interpretation, a phenomenon known as the modifiable areal unit problem [2–4]. To address these challenges, geostatistical techniques have been developed to filter the noise and downscale areal data to finer spatial scales [5].

One widely used geostatistical technique for smoothing or noise-filtering is Poisson kriging (PK). This technique was originally developed for characterizing the spatial distribution of rare wild species [6] from infrequent sightings and heterogeneous observation efforts (i.e., the total number of hours spent observing at a given location). Such data often feature a strongly skewed distribution with a large percentage of zeros and a few extremely large values. Since then, PK has been applied to a wide range of health outcomes

and spatial scales. For example, [7] mapped incidence rates of cholera and dysentery and associated measure of reliability in a 184 km$^2$ endemic area of Bangladesh using PK and data collected at the household level. Another study [8] applied PK to filter noise attached to lung and cervix cancer mortality rates recorded for white females in two contrasted county geographies: (1) state of Indiana that consists of 92 counties of fairly similar size and shape, and (2) four states in the western US (Arizona, California, Nevada and Utah) forming a set of 118 counties that are vastly different geographical units. In the western US and Utah, PK was used to smooth county-level incidence rates of drug poisoning deaths, populate data gaps and improve the reliability of rates recorded in sparsely populated counties [9].

Area-to-point (ATP) kriging [10,11] is a variant of kriging that allows the mapping of attribute values within each sampled geographical unit; in other words, converting choropleth maps into isopleth maps. This spatial downscaling, or disaggregation, is accomplished under the constraint that the average of point estimates returns the areal data (coherency constraint). The so-called pycnophilactic property, however, can lead to unrealistic estimates, such as negative values. For instance, the disaggregation of null areal data requires a mixture of positive and negative point estimates to ensure that the average within each unit is zero. This is especially problematic when implemented under the PK framework for analyzing health outcomes with a lot of zeros. A good example are COVID-19 incidence rates, as many geographical units had no cases detected at the beginning of the pandemic. This issue was, however, never mentioned in the studies that applied Poisson kriging to this type of data [12–14].

One quick and easy way to correct for negative ATP kriging estimates is to reset these estimates to zero [15], albeit at the cost of losing the pycnophilactic property. More elaborate solutions have been proposed in the geostatistical literature, such as (i) constraints on the kriging weights, (ii) the soft-kriging approach, and (iii) constrained predictions via non-linear optimization techniques. These solutions are reviewed in detail in [16]. Indicator kriging (IK) is explored here as an alternative to PK for filtering noise in data that display varying levels of reliability and a strongly skewed distribution with a large percentage of zeros and a few extremely large values.

This is not the first comparison of IK and PK performances. The authors of [17] compared IK and area-to-point PK for mapping patterns of herbivore species abundance in Kruger National Park, South Africa. This study indicated that IK was less accurate than PK for mapping animal abundance, in particular, the few large counts. Opposite conclusions were reached by [18], who investigated the applicability of ordinary kriging (OK), PK and IK to recreational fishery data, which are highly skewed, zero-inflated and when expressed as catch rates are impacted by the small number problem. In that case, IK was found to provide more accurate predictions of the latent catch rate for the three fish species under study compared to OK and PK, the later generating smoother spatial distributions. The present comparison study however differs in two major aspects: (1) the spatial support of the data are not regular squares but municipalities with irregular shape and size, and (2) the uncertainty attached to the observations is accounted for through the "soft" indicator coding (i.e., indicators are valued between zero and one) of a binomial distribution defined by the incidence rate and the underlying population size instead of a "hard" (i.e., indicators are zero or one) indicator coding of original data [15].

In this paper, we present an application of geostatistical techniques for the smoothing and downscaling of areal data, with a focus on disease incidence rates. We illustrate the advantages and limitations of PK and compare its performance to an innovative implementation of IK that propagates the uncertainty attached to the data through the creation of indicator transforms. The study is illustrated using daily COVID-19 incidence rates recorded in 2020–2021 for each of the 581 municipalities in Belgium.

## 2. Materials and Methods

### 2.1. COVID-19 Incidence Data

　　The geostatistical mapping approach is illustrated using daily COVID-19 incidence rates recorded in 2020–2021 for each of the 581 municipalities in Belgium (300 in the Flemish, 262 in the Walloon and 19 in the Brussels-Capital Region); see Figure 1A. The data were collected by Sciensano, the Belgian institute for health [19]. No individual-level information (e.g., gender, age) was available. Each rate was calculated as the sum of newly confirmed positive cases in the past 7 days divided by the population size of the respective municipality as of 1 January 2020 according to Statbel, the Belgian statistical office (Figure 1B). Rates are expressed as number of cases per 10,000 inhabitants. Belgian municipalities have an average area of 52.8 km$^2$, and finer details on the spatial distribution of the population was provided by a raster (31,557 cells of 1 km$^2$ each) of the 2016 population (Figure 1C). These high-resolution population data were derived through the geocoding of individual addresses from the National Register of Natural Persons (RNPP) and are publicly available [20]. Only very few individuals could not be geolocated (6827 individuals out of a population of 11,492,641 in 2020), leading to very accurate statistics.

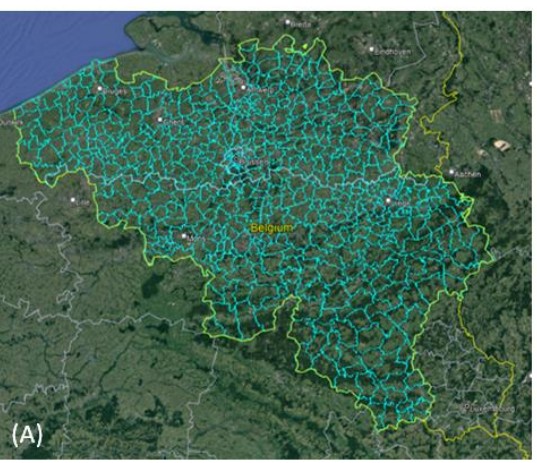

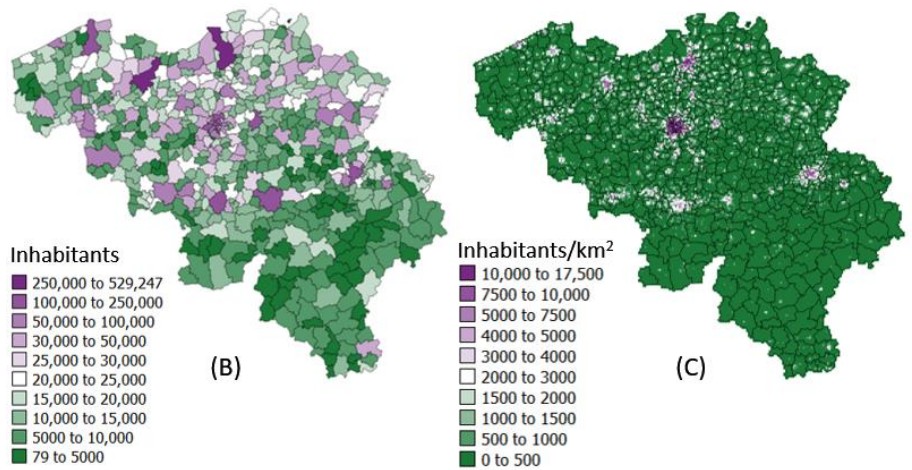

**Figure 1.** (**A**) Google Earth map overlaid with boundaries of 581 municipalities in Belgium. Population size recorded for: (**B**) each municipality as of 1 January 2020 and (**C**) 31,557 cells of 1 km$^2$ each in 2016, providing details about the spatial distribution of population with each administrative unit.

*2.2. Poisson Kriging*

Let $\{z(v_\alpha,t), \alpha = 1, \ldots, M\}$ be the set of COVID-19 incidence rates (areal data) recorded at M = 581 municipalities $v_\alpha$ on day t. Since the same analysis is undertaken independent-ly for every day t, the temporal reference is omitted from the notation hereafter for simplicity. Each rate is calculated as $z(v_\alpha) = d(v_\alpha)/n(v_\alpha)$, where $d(v_\alpha)$ is the number of positive cases and $n(v_\alpha)$ is the size of the population at risk (i.e., total population of the municipality). The objective of the analysis is twofold: (1) filter the noise attached to the observed rates $z(v_\alpha)$, and (2) estimate incidence rates at L = 31,557 nodes $u_l$ of a 1 km spacing grid discretizing the entire country of Belgium (spatial disaggregation). The results will be two sets of rate estimates for any single day *t*: $\{\hat{z}_{PK}(v_\alpha), \alpha = 1, \ldots, M\}$ and $\{\hat{z}_{PK}(u_l), l = 1, \ldots, L\}$ with the following coherency constraint that is satisfied for each municipality $v_\alpha$:

$$\hat{z}_{PK}(v_\alpha) = \frac{1}{n(v_\alpha)} \sum_{l=1}^{L_\alpha} n(u_l)\hat{z}_{PK}(u_l) \text{ with } n(v_\alpha) = \sum_{l=1}^{L_\alpha} n(u_l) \tag{1}$$

where $L_\alpha$ is the number of grid cells within municipality $v_\alpha$ and $n(u_l)$ represents the population within the grid cell centered on the node $u_l$. In both cases, each rate estimate is calculated as the weighted sum of rates recorded in $(B - 1)$ neighboring municipalities, besides the municipality $v_\alpha$ where the estimation is taking place:

$$\hat{z}_{PK}(v_\alpha) = \sum_{\beta=1}^{B} \lambda_\beta z(v_\beta) \text{ and } \hat{z}_{PK}(u_l) = \sum_{\beta=1}^{B} \kappa_\beta z(v_\beta) \tag{2}$$

where the weights are the solution of the following systems of linear equations known respectively as area-to-area (ATA) PK and area-to-point (ATP) PK:

$$\sum_{\beta=1}^{B} \lambda_\beta \left[ \overline{C}_R(v_\beta, v_\theta) + \delta_{\beta\theta} \frac{\overline{z}}{n(v_\theta)} \right] + \mu(v_\alpha) = \overline{C}_R(v_\alpha, v_\theta) \ \theta = 1, \ldots, B$$

$$\sum_{\beta=1}^{B} \lambda_\beta = 1 \tag{3}$$

$$\sum_{\beta=1}^{B} \kappa_\beta \left[ \overline{C}_R(v_\beta, v_\theta) + \delta_{\beta\theta} \frac{\overline{z}}{n(v_\theta)} \right] + \mu(u_l) = \overline{C}_R(u_l, v_\theta) \ \theta = 1, \ldots, B$$

$$\sum_{\beta=1}^{B} \kappa_\beta = 1, \tag{4}$$

where μ (.) is a Lagrange multiplier accounting for the unit sum constraint on the weights. The term $\overline{z}$ is the population-weighted average of the M observed rates, and its division by $n(v_\theta)$ is an error variance term, which is added to the variance $\overline{C}_R(v_\theta, v_\theta)$, as the Kronecker delta $\delta_{\beta\theta}$ is 1 if $\beta = \theta$ and 0 otherwise. Thus, municipalities with a small population size $n(v_\theta)$ (i.e., small denominator) receive smaller kriging weights, as incidence rates based on fewer cases are viewed as more error-prone (small number problem).

Under the assumption of second-order stationarity, the area-to-area covariance $\overline{C}_R(v_\beta, v_\theta)$ is numerically approximated as the average of the point-support covariance C (**h**) calculated between any two locations discretizing the geographical units corresponding to municipalities $v_\beta$ and $v_\theta$. Similarly, the area-to-point covariance $\overline{C}_R(u_l, v_\theta)$ is estimated by averaging the covariance C (**h**) computed between grid node $u_l$ and a set of locations discretizing the geographical unit $v_\theta$.

The point-support covariance C (**h**) is inferred in three steps. First, an area-based semivariogram is calculated from incidence rates using the Poisson estimator introduced in [5]:

$$\hat{\gamma}_v(\boldsymbol{h}) = \frac{0.5}{\sum_{\alpha,\beta}^{N(\boldsymbol{h})} \frac{n(v_\alpha) \times n(v_\beta)}{n(v_\alpha) + n(v_\beta)}} \sum_{\alpha,\beta}^{N(\boldsymbol{h})} \left\{ \frac{n(v_\alpha) \times n(v_\beta)}{n(v_\alpha) + n(v_\beta)} \left[ z(v_\alpha) - z(v_\beta) \right]^2 - \bar{z} \right\}, \tag{5}$$

where N (**h**) is the number of pairs of municipalities ($v_\alpha$, $v_\beta$) whose population-weighted centroids are separated by the vector **h**, and $\bar{z}$ is the population-weighted mean of the M incidence rate. The squared spatial increments $\left[ z(v_\alpha) - z(v_\beta) \right]^2$ are weighted by a function of their respective population sizes, $n(v_\alpha) \times n(v_\beta)/[n(v_\alpha) + n(v_\beta)]$, a term which is inversely proportional to their standard deviations [6]. More importance is thus given to the more reliable data pairs (i.e., smaller standard deviations). Second, a model is fitted to the experimental semivariogram $\hat{\gamma}_v(\boldsymbol{h})$ and deconvoluted using an iterative procedure [21] to derive the point-support semivariogram model $\gamma(\boldsymbol{h})$. Last, the covariance is calculated as C(h) = C (0) $-$ $\gamma(\boldsymbol{h})$, where C (0) is the sill of the semivariogram $\gamma(\boldsymbol{h})$.

*2.3. Soft Indicator Kriging*

To avoid the issue of negative estimates, $\hat{z}_{PK}(\boldsymbol{u}_l) < 0$, being generated during the disaggregation of municipality-level incidence rates by PK, a novel methodology was implemented. It relies on a soft indicator coding of rate data, followed by their spatial disaggregation using ATP kriging [11]. In the case of daily incidence rates, this new approach proceeds for each day as follows:

1.  Compute K = 50 percentiles $z_k$ of the frequency distribution of M = 581 municipality rates, F(.), as: $z_k = F^{-1}(p_k)$ with $\left\{ p_k = p_{min} + (k-1) \times \frac{(1 - p_{min})}{50}, k = 1, \cdots, K \right\}$ where $p_{min} = F(z_{min})$ is the proportion of rates no greater than the minimum observed rate $z_{min}$. This formulation avoids obtaining a series of zero-valued thresholds for days where no cases were recorded in many municipalities.

2.  For each municipality $v_\alpha$:

    *   Create a binomial distribution Bi ($n(v_\alpha)$,$z(v_\alpha)$) characterized by the daily incidence rate $z(v_\alpha)$ and the population $n(v_\alpha)$ within that geographical unit. This step allows one to capture the uncertainty attached to the observed rate $z(v_\alpha)$, which can be substantial for municipalities that are sparsely populated (i.e., small population size $n(v_\alpha)$),

    *   Discretize the probability distribution using the set of K thresholds $z_k$ calculated at step 1: $\{ j(v_\alpha; z_k) = F_{Bi}(v_\alpha; z_k), k = 1, \cdots, K \}$ where $F_{Bi}(v_\alpha)$ is the cumulative binomial distribution for the $\alpha$-th municipality. The quantity $j(v_\alpha; z_k)$ represents the probability that the underlying rate is no greater than the threshold $z_k$ for municipality $v_\alpha$.

3.  For each threshold $z_k$:

    *   Calculate and model the population-weighted indicator semivariogram as:

$$\gamma(\boldsymbol{h}; z_k) = \frac{1}{\sum_{\alpha,\beta}^{N(\boldsymbol{h})} n(v_\alpha) \times n(v_\beta)} \sum_{\alpha,\beta}^{N(\boldsymbol{h})} n(v_\alpha) \times n(v_\beta) \left[ j(v_\alpha; z_k) - j(v_\beta; z_k) \right]^2$$

    where N (**h**) is the number of pairs of municipalities ($v_\alpha$, $v_\beta$) whose population-weighted centroids are separated by the vector **h** [19].

    *   Use this model and ATP ordinary kriging to disaggregate the probabilities $j(v_\alpha; z_k)$ (i.e., soft indicator data) at the nodes $\boldsymbol{u}_l$ of a 1 km spacing grid discretizing the country (total number of nodes is L = 31,557).

4.  For each node $u_l$ of the discretization grid:
    - Assemble the K estimated probabilities $j^*(u_l; z_k)$ into a probability distribution.
    - Correct for order relation deviations [15] as: (1) each probability $j^*(u_l; z_k)$ can be negative or larger than 1 since it was estimated by ATP kriging (same potential issues as PK), and (2) the set of K probabilities $j^*(u_l; z_k)$ were estimated separately, with no guarantee that $j^*(u_l; z_k) \leq j^*(u_l; z_{k+1}) \, \forall \, z_{k+1} > z_k$.
    - Create a continuous distribution using linear interpolation between thresholds, as well as between the first (last threshold) and the minimum (maximum) observed rate.
    - Calculate the mean $\hat{z}_{IK}(u_l)$ and variance of the local probability distribution (ccdf).

5.  For each municipality $v_\alpha$:
    - Estimate each of the K probabilities $j^*(v_\alpha; z_k)$ as the population-weighted average of the corresponding probabilities $j^*(u_l; z_k)$ for all nodes $u_l$ that fall within that geographical unit, i.e.,

$$j^*(v_\alpha; z_k) = \frac{1}{n(v_\alpha)} \sum_{l=1}^{L} i(u_l; v_\alpha) \times n(u_l) \times j^*(u_l; z_k) \tag{6}$$

    with $i(u_l; v_\alpha) = 1$ if $u_l \in v_\alpha$ and 0 otherwise. $n(u_l)$ is the population size within the raster cell centered on node $u_l$ while $n(v_\alpha)$ is the population for the $\alpha$-th municipality. Each probability $j^*(v_\alpha; z_k)$ can be interpreted as the population-weighted fraction of the $\alpha$-th municipality area, $\mathcal{A}_\alpha$, where the threshold $z_k$ is not exceeded.
    - Assemble the K estimated probabilities $j^*(v_\alpha; z_k)$ into a probability distribution.
    - Correct for order relation deviations and create a continuous distribution using linear interpolation between thresholds, as well as between the first (last threshold) and the minimum (maximum) observed rate.
    - Calculate the mean $\hat{z}_{IK}(v_\alpha)$ and variance of the local probability distribution (ccdf).

*2.4. Software*

The analysis was conducted using the following public-domain software: (1) SpaceStat 4.0 [22] for Poisson kriging, ATP kriging and some of the data processing (e.g., creation of interpolation grids, mapping), and (2) program AUTO-IK [23] for indicator kriging.

**3. Results**

*3.1. Temporal Trend and Spatial Patterns*

The proposed approach (IK) was compared to results obtained by both ATA (area-to-area) PK for municipalities and ATP (area-to-point) PK for grid nodes. The comparison was conducted for four dates of the pandemic; from the time series of 365 daily incidence rates starting on 1 March 2020 (Figure 2), the analysis was conducted for times t = 10 days (10 March 2020), t = 200 days (16 September 2020), t = 242 days (28 October 2020) and t = 300 days (25 December 2020). This choice allowed one to consider a range of frequency distributions, from low incidence rates (mean = 0.444 cases/10,000 inhabitants) with 64% of null rates (10 March 2020), to a date (25 December 2020) with an average incidence rate of 11.66 cases/10,000 inhabitants and only a few municipalities with no COVID-19 cases; see Table 1. The peak was reached on 28 October 2020, when the incidence rate per municipality ranged between 12.98 and 303.8 cases/10,000 inhabitants, with a mean of 105.63. Note that at the beginning of the pandemic, the testing capacity was low, so that only patients admitted to the hospital were tested; the rates were therefore largely underestimated as for other infectious diseases [24]. This is visible in the amplitude of the first peak (t = 40 days),

which is 10 times smaller than the second peak (t = 242 days), while mortality rates were similar [25].

**Table 1.** Summary statistics of COVID-19 incidence rates (number of cases per 10,000 inhabitants) recorded at four different dates for all 581 Belgian municipalities.

| Statistics | Dates | | | |
|---|---|---|---|---|
| | **10 March 2020 (t = 10)** | **16 September 2020 (t = 200)** | **28 October 2020 (t = 242)** | **25 December 2020 (t = 300)** |
| Mean | 0.444 | 6.049 | 105.6 | 11.66 |
| Variance | 0.932 | 31.97 | 3573 | 55.65 |
| Minimum | 0.0 | 0.0 | 12.98 | 0.0 |
| Maximum | 6.981 | 45.59 | 303.8 | 50.84 |
| % null values | 64.4 | 10.8 | 0.0 | 1.9 |

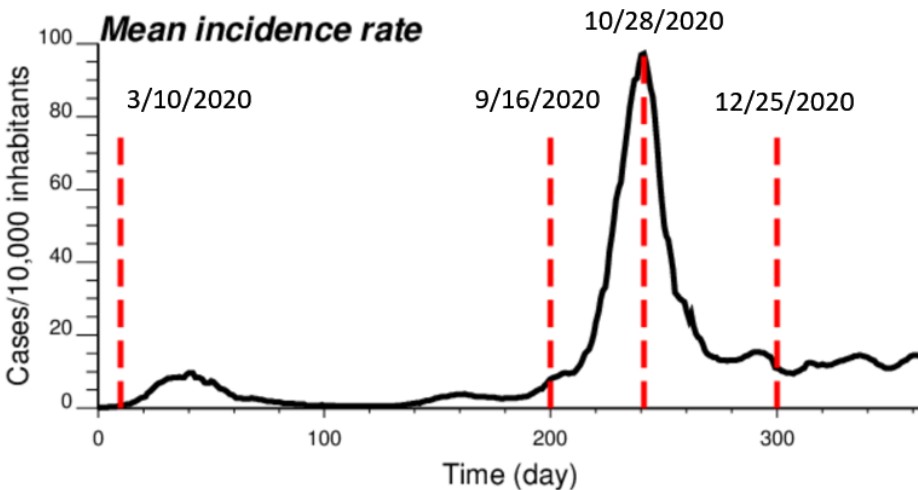

**Figure 2.** Time series of COVID-19 incidence rates (number of cases per 10,000 inhabitants), which were averaged over all 581 municipalities in Belgium. The time axis represents the number of days after 1 March 2020, while vertical red lines denote the four dates used for the comparison study.

The spatial distribution of incidence rates recorded at the municipality level was mapped in Figure 3 for each of the four dates. As the pandemic progressed and the incidence rate increased, a spatial pattern of higher incidence in the southeastern part of Belgium emerged. This spatial structure was particularly pronounced at the peak of the pandemic (28 October 2020). The semivariograms in Figure 4 confirmed this visual interpretation; in particular, the long-range structure became predominant while the relative nugget effect (i.e., discontinuity at the origin expressed as proportion of the sill) vanished as the infectious disease spread.

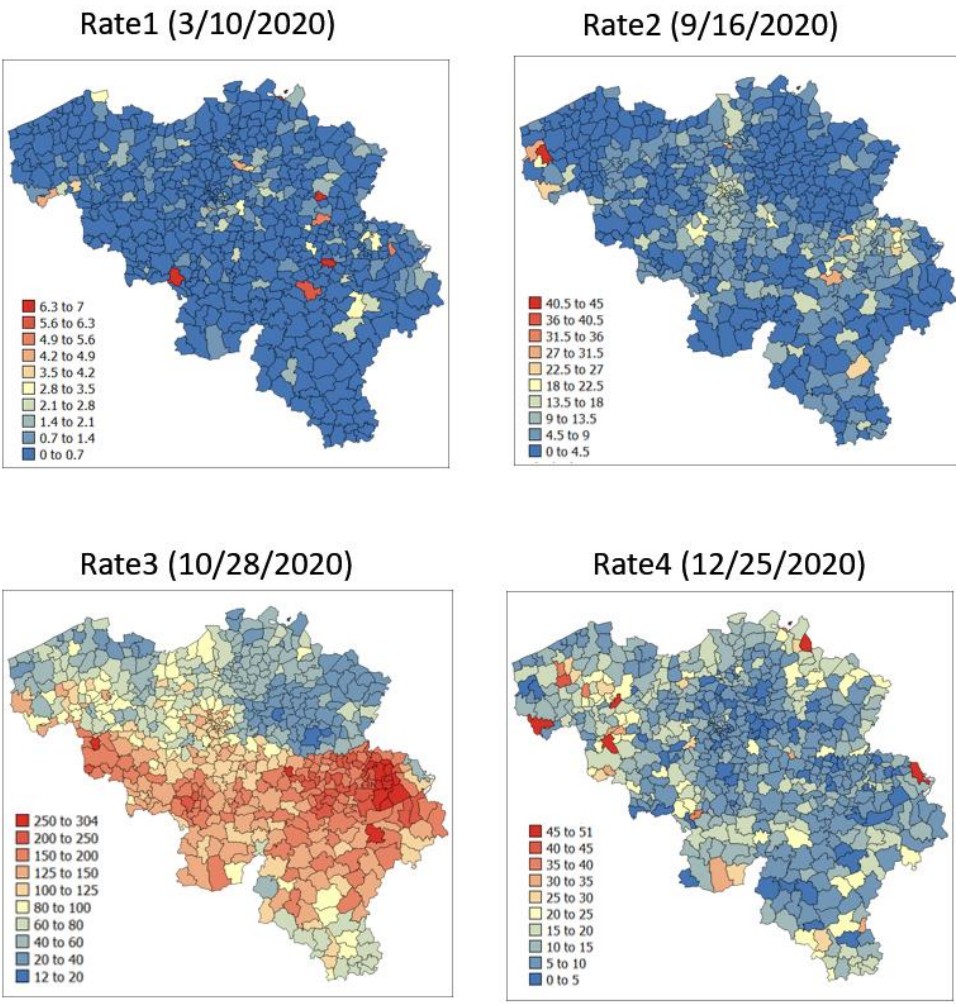

**Figure 3.** Maps of municipality-level COVID-19 incidence rates (number of cases per 10,000 inhabitants) recorded at four different dates.

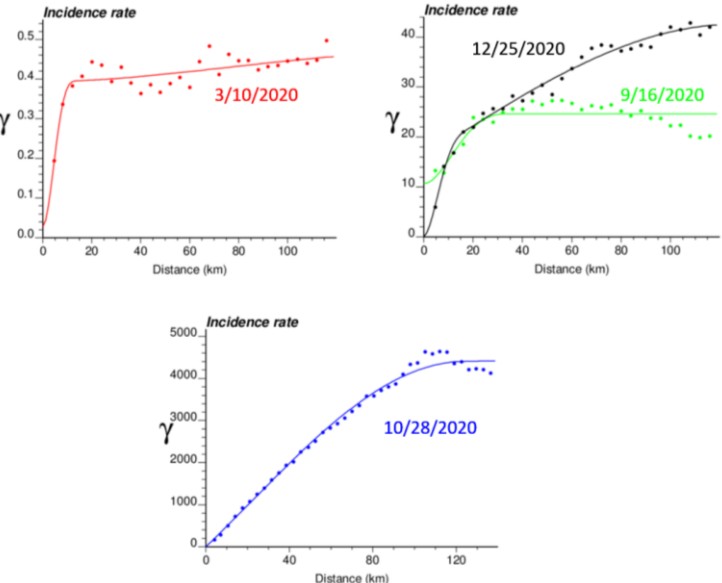

**Figure 4.** Experimental Poisson semivariogram calculated from municipality-level COVID-19 incidence rates (number of cases per 10,000 inhabitants) recorded at four different dates, with the model fitted.

Similar results were obtained in studies on the temporal variation of spatial autocorrelation of COVID-19 cases. In particular, using county-level incidence rates in Poland, ref. [14] found that the overwhelming majority of dates without any spatial autocorrelation, as measured by the Poisson semivariogram, happened at the beginning of the pandemic (March to August 2020) when the number of cases was low, and geographical units with extremely low and high values were directly adjacent to each other. Other measures of autocorrelation, such as Global Moran's I or the spatial component of generalized linear models, revealed similar temporal trends for other countries [26–28].

The application of IK required the computation and modelling of semivariograms, albeit in a much larger number (50 vs. 1), which was conducted automatically by the program AUTO-IK [23]. Indicator semivariograms for the first (second for 28 October 2020) threshold were plotted for all four dates (Figure 5), while Figure 6 shows experimental and modeled semivariograms for three additional thresholds (# 10, 25, 40) on two different dates (10 March 2020, 25 December 2020). Like for Poisson semivariograms (Figure 4), stronger spatial structures, in particular longer ranges of autocorrelation, were observed at later dates, as incidence rate increased.

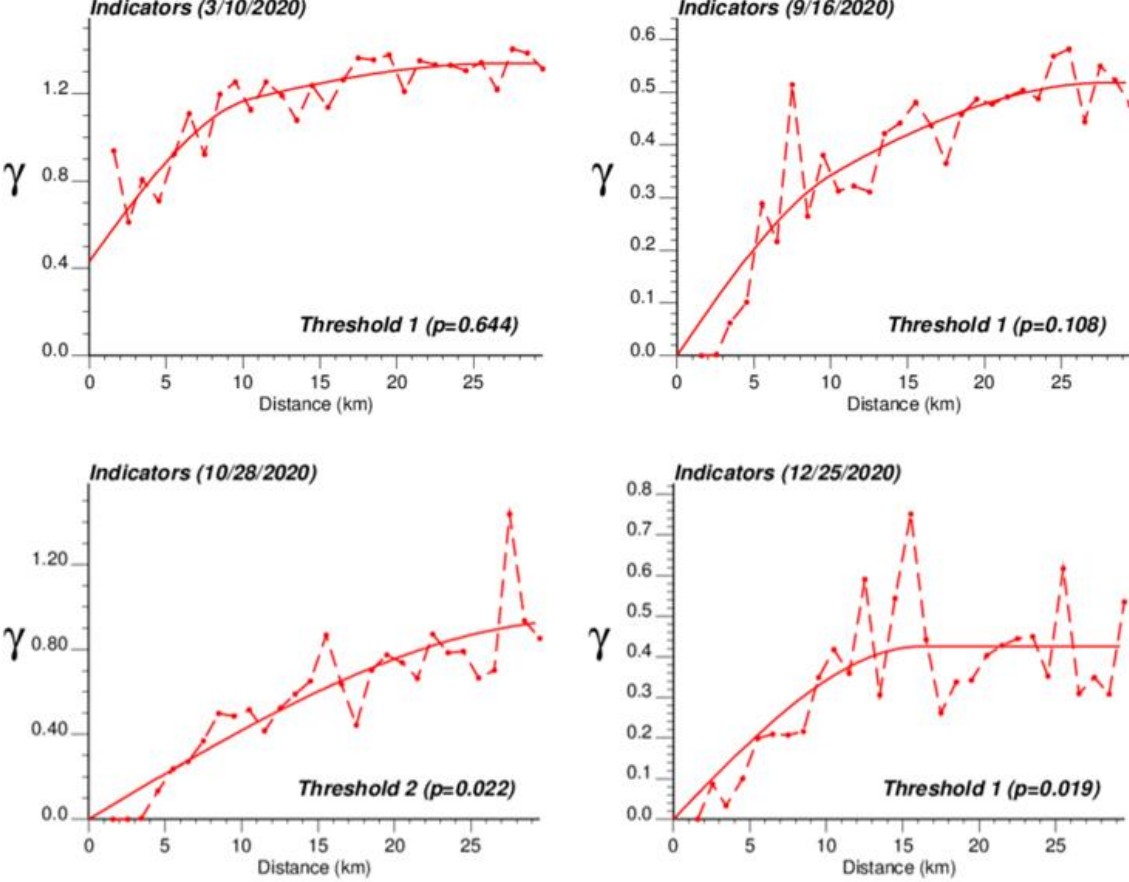

**Figure 5.** Experimental population-weighted indicator semivariogram calculated for the first (second for 28 October 2020) threshold at four different dates, with the model fitted. The first threshold corresponds to a null COVID-19 incidence rate that was recorded among 1.9% (25 December 2020) to 64.4% (10 March 2020) of Belgian municipalities. COVID-19 cases were identified in all 581 Belgian municipalities on 28 October 2020 when the mean incidence rate peaked.

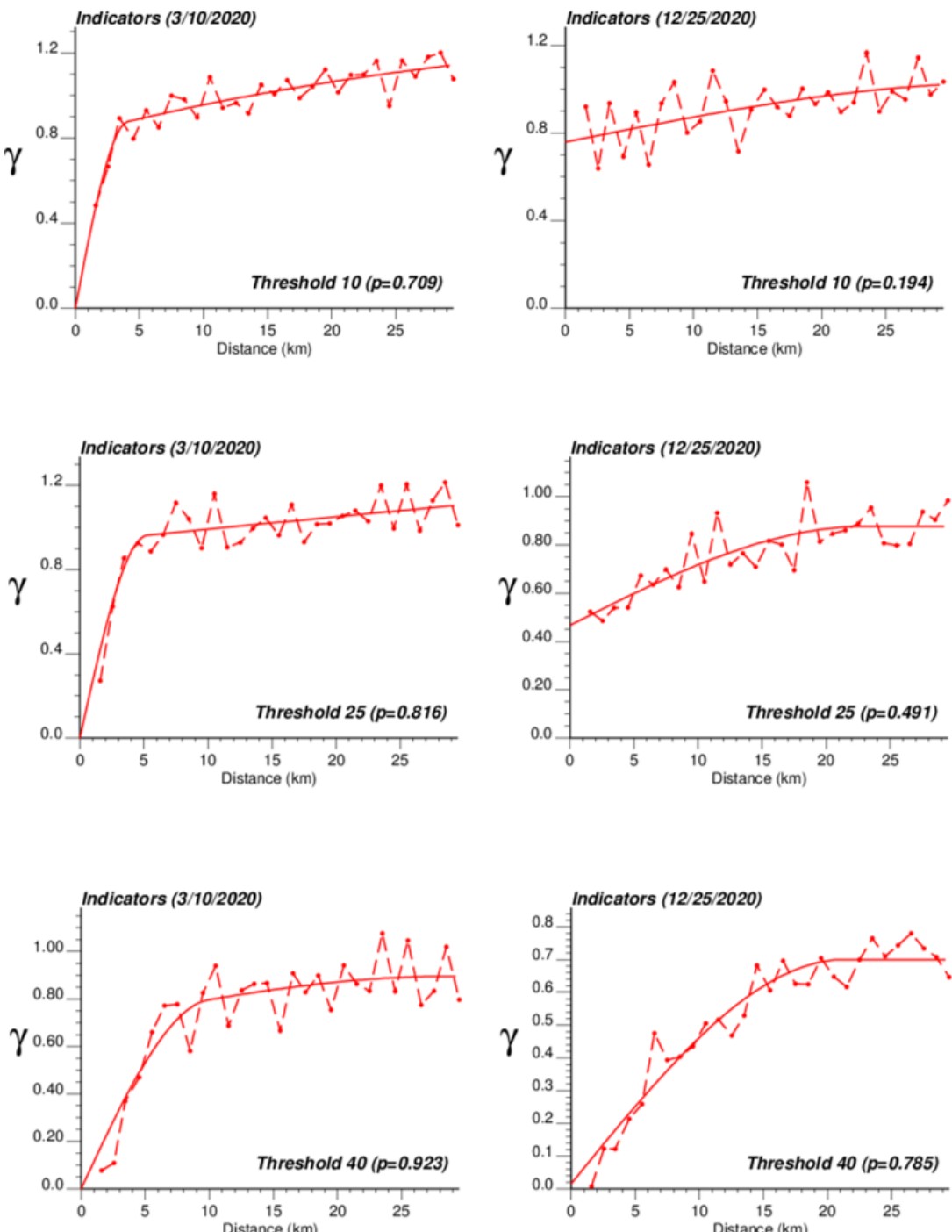

**Figure 6.** Experimental population-weighted indicator semivariogram calculated for three different thresholds (# 10, 25, 40) and two different dates (10 March 2020, 25 December 2020), with the model fitted.

### 3.2. Kriging Estimates

The semivariogram models in Figure 4 were used to filter the noise attached to municipality-level rates using area-to-area PK (ATAPK) and the eight closest municipalities (*B* = 8 in Equation (2)). ATA kriging results were mapped for the first and last dates at the top of Figures 7 and 8, while statistics (variance, rank correlation with observed rates) for all four dates were listed in the first two rows of Tables 2 and 3. Compared to PK, the IK approach generated less smoothing, both at the municipality level (compare top maps

in Figures 7 and 8) and after downscaling to a 1 km resolution (compare middle maps in Figures 7 and 8).

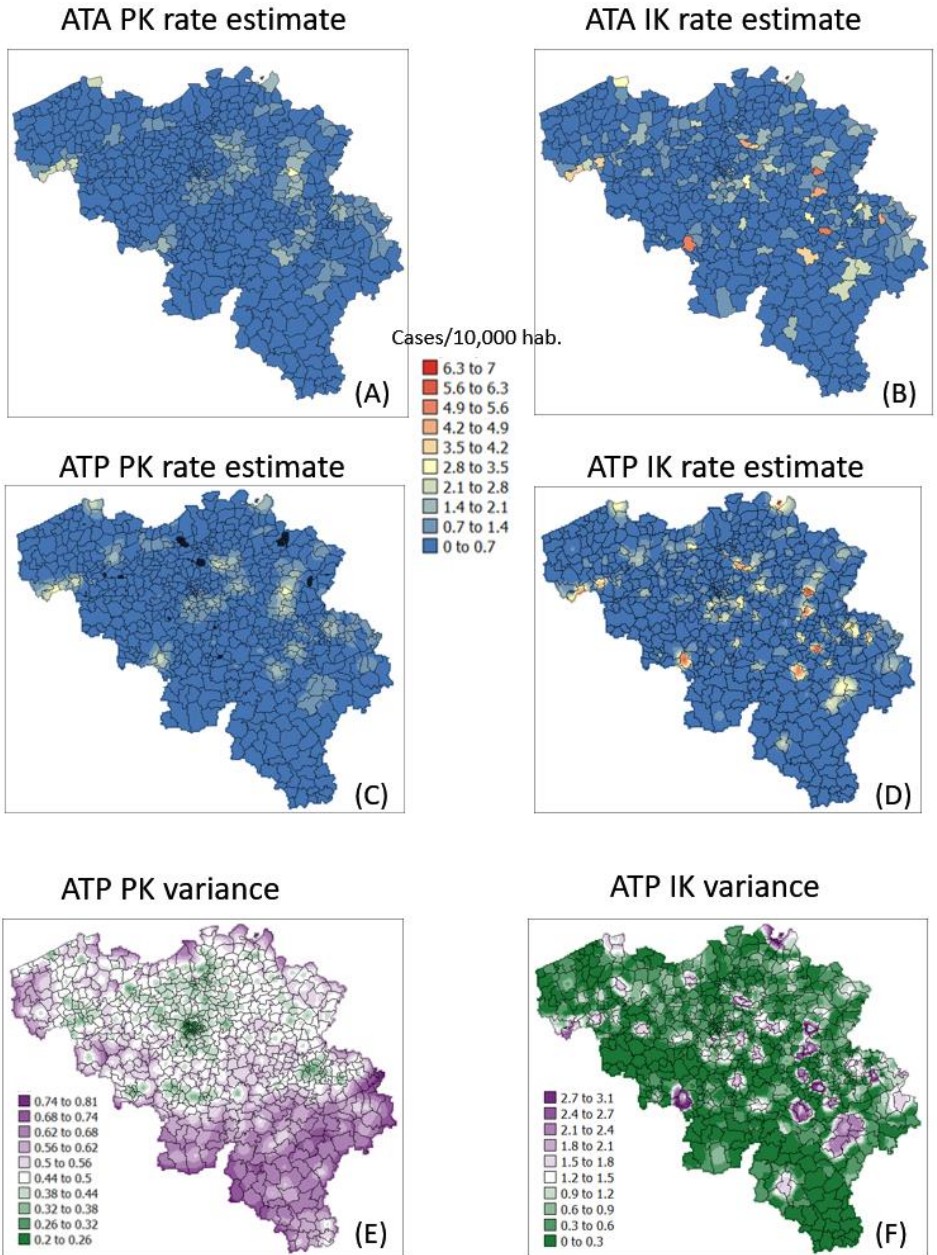

**Figure 7.** Maps of smoothed COVID-19 incidence rates (number of cases per 10,000 inhabitants) calculated by PK and IK at the municipality level (ATA kriging, **A**,**B**) and at the nodes of a 1 km spacing grid (ATP kriging, **C**,**D**) for time period 1 (10 March 2020). Black pixels indicate negative kriging estimates. Bottom maps (**E**,**F**) show the kriging variance for ATP kriging.

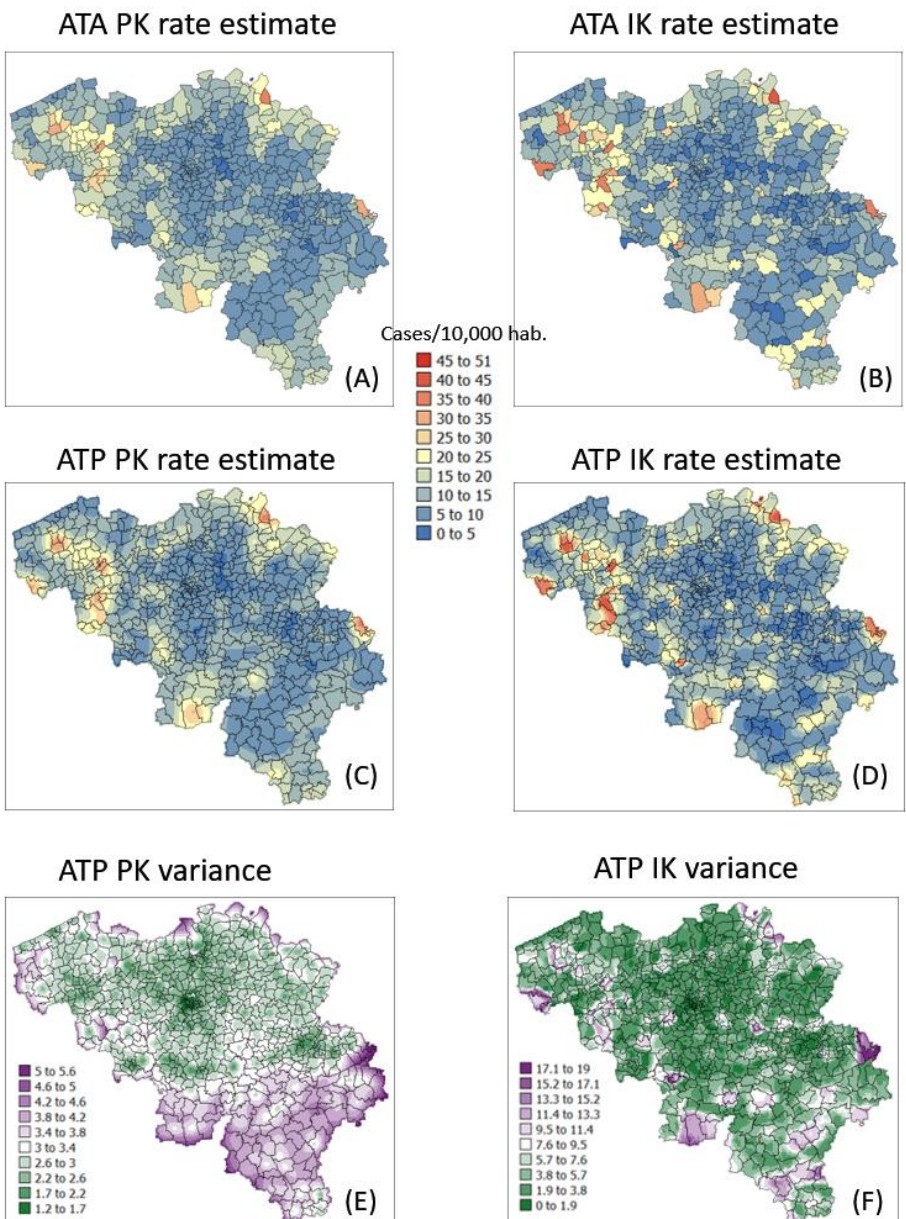

**Figure 8.** Maps of smoothed COVID-19 incidence rates (number of cases per 10,000 inhabitants) calculated by PK and IK at the municipality level (ATA kriging, **A**,**B**) and at the nodes of a 1 km spacing grid (ATP kriging, **C**,**D**) for time period 1 (25 December 2020). Black pixels indicate negative kriging estimates. Bottom maps (**E**,**F**) show the kriging variance for ATP kriging.

**Table 2.** Variance of Poisson (PK) and indicator kriging (IK) estimates calculated at the municipality level (ATA) and after spatial disaggregation (ATP) for four different dates.

| | Dates | | | |
|---|---|---|---|---|
| **Variance** | **10 March 2020 (t = 10)** | **16 September 2020 (t = 200)** | **28 October 2020 (t = 242)** | **25 December 2020 (t = 300)** |
| ATA PK | 0.194 | 15.91 | 3219 | 25.10 |
| ATA IK | 0.754 | 25.77 | 3575 | 49.07 |
| ATP PK | 0.190 | 10.49 | 3031 | 26.95 |
| ATP IK | 0.559 | 16.73 | 3225 | 45.81 |

**Table 3.** Rank correlation between Poisson (PK) and indicator kriging (IK) estimates calculated at the municipality level (ATA) and after spatial disaggregation (ATP) for four different dates. At the municipality level, the rank correlation with observed rates is also listed.

| | Dates | | | |
|---|---|---|---|---|
| **Correlation** | **10 March 2020 (t = 10)** | **16 September 2020 (t = 200)** | **28 October 2020 (t = 242)** | **25 December 2020 (t = 300)** |
| ATA PK vs. rate | 0.645 | 0.846 | 0.989 | 0.851 |
| ATA IK vs. rate | 0.883 | 0.976 | 0.998 | 0.997 |
| PK vs. IK (ATA) | 0.645 | 0.889 | 0.989 | 0.862 |
| PK vs. IK (ATP) | 0.849 | 0.875 | 0.980 | 0.907 |

The magnitude of the smoothing effect, hence discrepancies between the two approaches, was the largest when the average incidence rate was low, and many municipalities had no cases. For example, the variance of noise-filtered municipality-level rates is 0.194 (PK) vs. 0.754 (IK) for the first time period (10 March 2020) with 64.4% of null rates, while the ratio of variances is the smallest (3219 vs. 3575) at the peak of the pandemics when cases were recorded in all municipalities (Table 3). Similarly, the correlation between the two sets of kriged rates strengthened as we moved from time period 1 to 3: r = 0.645 to 0.989 (Table 3).

The smaller variance of PK estimates at t = 10 days is caused by the weaker spatial structure (shorter range and larger nugget effect of semivariograms in Figure 4), which led to assigning almost as much weight to remote observations than closer ones in Equation (2). In other words, the kriging estimate starts mimicking the arithmetical average and deviates more from the original rate. This is reflected by the lower correlation between ATA PK rates and observed rates compared to ATA IK rates and observed rates, in particular for the first date (r = 0.883 for IK vs. 0.645 for PK); see Table 3 (first two rows). Interestingly, the greater variance of IK estimates vs. PK estimates is not observed over the entire range of values. The scatterplots of Figure 9 illustrate the narrower range exhibited by IK rate estimates vs. PK results for municipalities with no case (green ellipses). This is a desirable feature, as one should be suspicious of estimated rates that greatly exceed observed rates after smoothing.

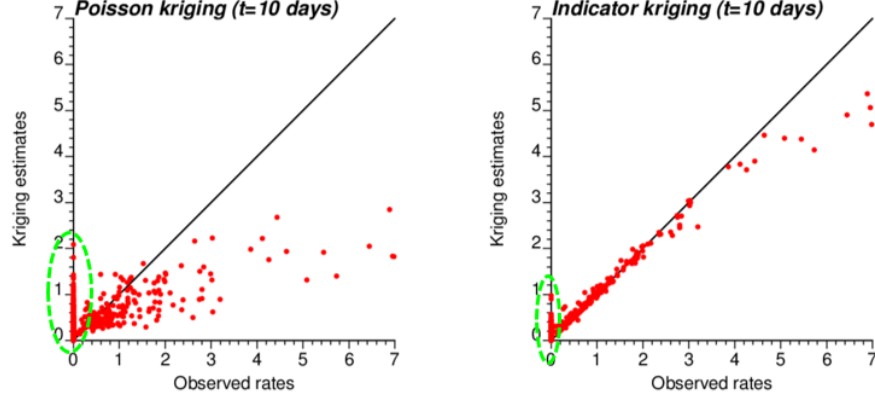

**Figure 9.** Scatterplots of COVID-19 incidence rates smoothed by PK and IK at the municipality level (ATA kriging) versus rates recorded for time period 1 (10 March 2020). Notice the larger smoothing effect (i.e., smaller spread) caused by PK vs. IK, except for null incidence rates where the estimates range from 0 to 2 cases per 10,000 inhabitants vs. 0 to 1 for IK (green ellipse).

Similar to the case of municipality rates, the correlation between the two sets of gridded estimates (ATP kriging) increases with time as the average incidence rate increases:

r = 0.849 to 0.980; see Table 3 (last row). In addition to reducing the smoothing effect (Table 2, last two rows), disaggregation by IK also avoids the generation of negative estimates for the first time period where many municipalities do not have any case; see black pixels in the map of Figure 7C. The disaggregation of municipality-level probabilities $\{j(v_\alpha; z_k), k = 1, \cdots, K\}$ using ATP ordinary kriging (Step 3 in Section 2.3) can, however, generate a set of probabilities $\{j^*(\boldsymbol{u}_l; z_k), k = 1, \cdots, K\}$ that violates order relations, including negative probabilities, and needs to be corrected. The magnitude of these corrections (Table 4, second line) is of the order of 0.045 and are less frequent at the peak of the pandemic (t = 242 days): 43% vs. 80% for other days (Table 4, first line), likely because of the absence of zero incidence rates. Once corrected at the grid level, no more order relation deviation is observed after aggregation to the municipality level; see Table 4 (last two lines).

**Table 4.** Frequency and magnitude of corrections applied to IK-based ccdfs that violate order relations at the grid node (ATP IK) and municipality (ATA IK) levels.

| | Dates | | | |
|---|---|---|---|---|
| **Order Relations** | **10 March 2020 (t = 10)** | **16 September 2020 (t = 200)** | **28 October 2020 (t = 242)** | **25 December 2020 (t = 300)** |
| Freq. (ATP) | 0.798 | 0.791 | 0.427 | 0.708 |
| Magn. (ATP) | 0.044 | 0.042 | 0.051 | 0.049 |
| Freq. (ATA) | 0.0 | 0.0 | 0.0 | 0.0 |
| Magn. (ATA) | 0.0 | 0.0 | 0.0 | 0.0 |

*3.3. Kriging Variances*

In addition to rate estimates, PK and IK differ in terms of the variance of prediction errors; see distinct spatial patterns at the bottom of Figures 7 and 8. Once again, differences are the largest for the first time period: the correlation between the two sets of ATA kriging standard deviations (not mapped) is only -0.176, while the correlation is 0.728 for the last time period (Table 4, first row). While the PK variance is a function of the geometric configuration of the data (i.e., larger variance where distance between municipality centroids is large) and the population size (i.e., smaller variance in more heavily populated municipalities), the variance of the IK-based probability distributions is also a function of the mean of these distributions (i.e., risk estimates), a phenomenon known as proportional effect. This is illustrated by the correlation coefficients listed in Table 5. For ATA IK, the correlation between kriging standard deviation and estimates ranges between 0.724 (t = 242) and 0.986 (t = 10) while it never exceeds 0.259 for PK. Similar results are obtained for ATP kriging (Table 5, bottom half) although the magnitude of negative and positive correlations is smaller, which is an artefact of the larger population size: n = 581 municipalities versus n = 31,557 grid nodes.

**Table 5.** Rank correlation between Poisson (PK) and indicator kriging (IK) outputs (i.e., estimated value, error standard deviation) calculated at the municipality level (ATA) and after spatial disaggregation (ATP) for four different dates. Correlation with municipality population size is also listed.

| | Dates | | | |
|---|---|---|---|---|
| **Correlation** | **10 March 2020 (t = 10)** | **16 September 2020 (t = 200)** | **28 October 2020 (t = 242)** | **25 December 2020 (t = 300)** |
| ATA | | | | |
| PKstd vs. IKstd | −0.176 | 0.379 | 0.728 | 0.721 |
| PKstd vs. PKest | −0.197 | −0.053 | 0.259 | 0.115 |
| IKstd vs. IKest | 0.986 | 0.657 | 0.724 | 0.619 |
| PKstd vs. Population | −0.908 | −0.933 | −0.980 | −0.919 |
| IKstd vs. Population | 0.175 | −0.339 | −0.737 | −0.689 |

**Table 5.** *Cont.*

| | Dates | | | |
|---|---|---|---|---|
| **Correlation** | **10 March 2020 (t = 10)** | **16 September 2020 (t = 200)** | **28 October 2020 (t = 242)** | **25 December 2020 (t = 300)** |
| ATP | | | | |
| PKstd vs. IKstd | −0.106 | 0.414 | 0.499 | 0.508 |
| PKstd vs. PKest | −0.208 | −0.040 | 0.257 | 0.116 |
| IKstd vs. IKest | 0.952 | 0.571 | 0.631 | 0.585 |
| PKstd vs. Population | −0.665 | −0.711 | −0.680 | −0.669 |
| IKstd vs. Population | 0.040 | −0.302 | −0.384 | −0.390 |

## 4. Conclusions

In this study, we addressed two common challenges associated with the analysis of health outcomes aggregated over small areas, namely filtering noise caused by small local population sizes and deriving estimates at different spatial scales. Geostatistical techniques, such as PK, have been widely used to address these challenges by accounting for the pattern of spatial correlation and neighboring observations in the smoothing and change in spatial support. However, a limitation of PK is the possibility of generating unrealistic rates that are either negative or larger than 100%.

To overcome this limitation, we proposed an alternative approach that relies on a soft indicator coding of probability distributions inferred from rate data and population size for a series of geographical units. The use of a binomial distribution was novel and allowed us to account for uncertainty attached to rates recorded in sparsely populated geographical areas. This was followed by a kriging interpolation to derive these distributions at different spatial scales. The effectiveness of this approach was demonstrated by applying it to daily COVID-19 incidence rates recorded in 2020–2021 for each of the 581 municipalities in Belgium.

The absence of data at the grid level prohibited the quantitative comparison of the accuracy of the two types of disaggregation procedures. The IK approach has, however, several attractive features: (1) the lack of negative kriging estimates, (2) the smaller smoothing effect, and (3) the better agreement with observed rates after aggregation, in particular, when the original rate was zero (i.e., epidemiologists might be suspicious of smoothed rates that deviate too much from observed rates). It is noteworthy that as the mean incidence rate increases, differences between approaches decrease, which can be viewed as a positive development.

Indicator kriging is computationally more expensive since it requires the solving of 50 kriging systems instead of a single PK system, yet it is still tractable as it took just a few CPU minutes in the present case. Another limitation is that probabilities estimated through disaggregation can be negative and might not form a valid probability distribution, with the need for a posteriori correction of these order relation deviations. Such correction has, however, been routinely applied in the geostatistical literature [29]. Future work will extend the applications of these techniques to chronic diseases, including the mapping of cancer incidence and mortality rates [30].

One of the main benefits of geostatistical techniques is their ability to incorporate multiple layers of secondary information that can be more densely sampled or available at a finer spatial resolution. This is particularly useful when disaggregating areal data to explain some of the unknown spatial variation present at local scales. One of the most widely applied technique is area-to-point residual kriging, where ancillary data are used to inform on within-area variation through a regression model, followed by the disaggregation of regression residuals by ATP kriging. This multivariate interpolation approach improved the accuracy of prediction in multiple studies, including the mapping of urban population density using remote sensing covariates [31], the prediction of soil organic carbon content [32] using recent soil measurements and disaggregated legacy

soil data (soil map) in addition to high resolution auxiliary variables (elevation, airborne radiometric K), and the downscaling of 25 km resolution surface soil moisture remote sensing products using fine scale auxiliary data [33]. Recent findings [34,35] indicate a connection between geo-environmental factors (e.g., air pollution, chemical exposures, climate and the built environment) and the transmission, susceptibility and severity of COVID-19. A similar methodology could thus be implemented for incorporating such environmental factors in the space-time prediction of COVID-19 incidence rates.

**Author Contributions:** Conceptualization, Pierre Goovaerts, Ellen Van De Vijver, Peter F. Goossens and Thomas Hermans; methodology, Pierre Goovaerts, Ellen Van De Vijver, Peter F. Goossens and Thomas Hermans; validation, Pierre Goovaerts, Ellen Van De Vijver, Peter F. Goossens and Thomas Hermans; formal analysis, Pierre Goovaerts; investigation, Pierre Goovaerts and Ellen Van De Vijver; resources, Pierre Goovaerts; writing—original draft preparation, Pierre Goovaerts; writing—review and editing, Pierre Goovaerts, Ellen Van De Vijver, Peter F. Goossens and Thomas Hermans; visualization, Pierre Goovaerts; supervision, Pierre Goovaerts and Ellen Van De Vijver; project administration, Ellen Van De Vijver; funding acquisition, Pierre Goovaerts and Ellen Van De Vijver. All authors have read and agreed to the published version of the manuscript.

**Funding:** This research was funded by grant R44CA132347-02 from the National Cancer Institute and grant G0G9820N of the FWO (Research Foundation—Flanders). The views stated in this publication are those of the author and do not necessarily represent the official views of the NCI.

**Data Availability Statement:** The data presented in this study are available on request from the corresponding author. Population data for each Belgium municipalities can be downloaded from https://statbel.fgov.be/sites/default/files/files/documents/bevolking/5.1%20Structuur%20van%20de%20bevolking/Bevolking_per_gemeente.xlsx, accessed on 9 May 2023. Population data at a finer spatial scale (1 km$^2$ grid cells) are available at https://statbel.fgov.be/en/open-data/population-according-km2-grid-2020, , accessed on 9 May 2023. A free 1-yr license of SpaceStat can be downloaded at https://biomedware.com/products/spacestat/, , accessed on 9 May 2023.

**Conflicts of Interest:** The authors declare no conflict of interest.

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
