# Peer review of "Comparison of Soft Indicator and Poisson Kriging for the Noise-Filtering and Downscaling of Areal Data: Application to Daily COVID-19 Incidence Rates"

_ijgi, doi:10.3390/ijgi12080328_

Round 1

Reviewer 1 Report

This well-written manuscript proposes a novel combination of indicator coding of disease rates obtained at an aggregate (municipality) level with area-to-point Indicator Kriging (IK) for smoothing and/or downscaling such aggregate rates at a 1km spatial resolution. It is posited that the resulting estimates do not suffer from the negative value effect, frequently encountered when downscaling disease rates with area-to-point Poisson Kriging (PK).

The mathematics appear correct and are kept at an appropriate level, the topic is of interest to the readership of this journal, and the results of the case study illustrate the applicability of the proposed method and its comparative performance to PK.

I therefore recommend that the manuscript be accepted for publication subject to revision.

General comments

It would be useful to discuss how one could avoid the problem of negative estimates in ATA or ATP PK. Are there workarounds? Please comment.

IK is not a perfect estimator. Order relation corrections are simply fixes for inconsistencies in estimated probabilities. IK-derived probabilities can frequently be negative or greater than 1. One simply translates the problem of negative rates estimates to that of negative estimated probabilities of exceeding a given threshold... This issue was simply hidden in the nice term “order relation deviations corrections”... It needs to be explicated and made clear to the readers... Without such ad hoc corrections the resulting mean of a local distribution could easily be negative as well...

Specific comments

How was the fine resolution population image of Figure 1C constructed? See also page 2 line 88. This input is critical to the entire procedure. Is there no uncertainty in such population estimates?

Page 5 line 147: is reference [5] indeed the one you intended to call for here? It seems that [5] deals with area-to-point PK not IK.

The fact that IK-derived probabilities of 1 km spatial resolution rates not to exceed a specific threshold within a geographical area, reproduce, when upscaled, the corresponding proportion of 1km rates no greater than that threshold for that geographical area (support) should be explicated.

Reference 8 is not defined on page 18 (line 405).

Reviewer 2 Report

The article addresses an interesting problem in kriging interpolation with discrete value data. It proposes a method for avoiding unrealistic estimates when the count is small and mostly zero. It seems that the proposed method yields better results compared to Poisson Kriging.

Apart from technical considerations, I think that the article might explain the usefulness of this work for the prediction (?) of covid cases. Since the actual data are known, the purpose of kriging analysis is not clear and whether it is related to some factors favorable for the spread of the disease.

Some minor remarks concern the following:

All figures show the municipalities and it is not clear to me how disaggregation to cells works

In figure 9 I cannot see the smaller spread caused by PK vs IK.

Reviewer 3 Report

The manuscript depicts and interesting topic and it is enough well written but needs some important improvements in the structure.

If authors well follow the suggestion the manuscript will have a serious chance to be suitable for pubblication and I will recommend this manuscript for pubblication for sure.

Firstly, I suggest you to improve the introduction better discussing the geostatistics applications in the health sector with particular regard to the use of Remote Sensing Data that permits to define also the spread risk factor through an analysis of the land cover types. To do so and help you I suggest to include the following references in your interesting work:

- https://doi.org/10.3390/life13040987

- https://doi.org/10.3390/rs15010178

- https://doi.org/10.3390/app13010390

In Material and Methods I suggest you to better discuss the software adopted and the steps followed.. Then please better describe how you got the data and how you treat them before performing the analysis you reported. How you treated biases?

Results are fine but please rporte EPSG or Reference System in each maps...

The discussion section is too short improve it. I strongly suggest to better discuss as future perspective of your study in light of the role that Earth Observation Data may have in epidemiology also the effect of climate change linked also to illnesses risks. To do so and help you, I suggest to include the following references:

- https://doi.org/10.1117/12.2533110

- https://doi.org/10.3390/cli9030047

Finally, include the conclusion section is totally missing....

Moderate editing of English language required.

Round 2

Reviewer 3 Report

The authors have improved significantly the manuscript. 

However, I still find some lacks in the introduction, especially concerning on the role that environmental patterns detected by remote sensing can have in the understanding of the zoonoses diseases like COVID-19. Therefore, to accomplish this part and improve the introduction I suggest the Authors to discuss this topic. in order to help you I suggest to consider the following paper:  https://doi.org/10.3390/life13040987; https://doi.org/10.1016/j.ijppaw.2015.01.006. 

Author Response

We acknowledge that the geostatistical prediction of COVID-19 incidence rates would benefit from the incorporation of geo-environmental factors that were recently identified in the literature. This development is discussed in a new paragraph at the end of the conclusions and the incorporation of 5 new references that were deemed relevant.